# MuSCLE: Multi Sweep Compression of LiDAR using Deep Entropy Models

**Sourav Biswas**[1,2]    **Jerry Liu**[1]    **Kelvin Wong**[1,3]    **Shenlong Wang**[1,3]    **Raquel Urtasun**[1,3]

[1]Uber Advanced Technologies Group    [2]University of Waterloo    [3]University of Toronto

`{souravb,jerryl,kelvin.wong,slwang,urtasun}@uber.com`

## Abstract

We present a novel compression algorithm for reducing the storage of LiDAR sensor data streams. Our model exploits spatio-temporal relationships across multiple LiDAR sweeps to reduce the bitrate of both geometry and intensity values. Towards this goal, we propose a novel conditional entropy model that models the probabilities of the octree symbols by considering both coarse level geometry and previous sweeps' geometric and intensity information. We then use the learned probability to encode the full data stream into a compact one. Our experiments demonstrate that our method significantly reduces the joint geometry and intensity bitrate over prior state-of-the-art LiDAR compression methods, with a reduction of 7–17% and 6–19% on the UrbanCity and SemanticKITTI datasets respectively.

## 1 Introduction

The past decade has witnessed numerous innovations in intelligent systems, thanks to an explosion of progress in sensing and AI algorithms. In particular, LiDAR sensors are extensively used in various applications such as indoor rovers, unmanned aerial vehicles, and self-driving cars to accurately capture the 3D geometry of the scene. Yet the rapid adoption of LiDAR has brought about a key challenge—dealing with the mounting storage costs associated with the massive influx of LiDAR data. For instance, a 64-line Velodyne LiDAR continuously scanning a given scene produces over *3 billion points* in a single hour. Hence, developing efficient and effective compression algorithms to store such 3D point cloud data streams is crucial to reduce the storage and communication bandwidth.

Unlike its well-studied image and video counterparts, point cloud stream compression is a challenging yet under-explored problem. Many prior approaches have focused on encoding a point cloud stream as independent *sweeps*, where each sweep captures a rough 360-degree rotation of the sensor. Early approaches exploit a variety of compact data structures to represent the point cloud in a memory-efficient manner, such as octrees [1], KD-trees [2], and spherical images [3]. More recent works along this direction utilize powerful machine learning models to encode redundant geometric correlations within these data structures for better compression [4, 5, 6]. In general, most of these aforementioned approaches do not make effective use of temporal correlations within point clouds. Moreover, these prior approaches have largely focused on compressing the geometric structure of the point cloud (the spatial coordinates); yet there has been little attention paid towards compression of other attributes, *e.g.* LiDAR intensity, which are crucial for many downstream tasks. Compressing such attributes along with geometric structure can make a significant impact on reducing storage.

In this paper, we present a novel, learning-based compression algorithm that comprehensively reduces the storage of LiDAR sensor data streams. Our method extends the recent success of octree-structured deep entropy models [6] for single LiDAR sweep compression to intensity-valued LiDAR streaming data. Specifically, we propose a novel *deep conditional entropy model* that models the probabilities of the octree symbols and associated intensity values by exploiting spatio-temporal correlations within the data: taking both coarse level information at the current sweep, as well as relevant neighboring

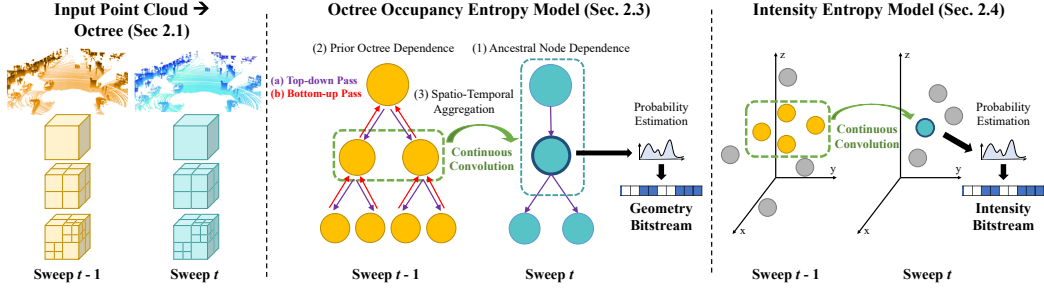

Figure 1: Comprehensive overview of our method. Our point cloud stream is serialized into an octree representation (Sec 2.1). We apply a spatio-temporal entropy model to the octree occupancy bytestream (Sec. 2.3), modeling ancestral dependence, prior octree dependence, and octree alignment. We also apply a deep entropy model to model the intensity stream (Sec. 2.4).

nodes information from the previous sweep. Unlike prior approaches, our method models the joint entropy across an entire point cloud sequence, while unifying geometry and attribute compression into the same framework.

We validate the performance of our approach on two large datasets, namely UrbanCity [7] and SemanticKITTI [8]. The experiments demonstrate that our method significantly reduces the joint geometry and intensity bitrate over prior state-of-the-art LiDAR compression methods, with a reduction of 7–17% on UrbanCity and 6–19% on SemanticKITTI. We also conduct extensive experiments showcasing superior performance against prior works on numerous downstream perception tasks.

## 2 Multi-Sweep LiDAR Compression

In this work, we propose a comprehensive framework for the *lossy* compression of LiDAR point cloud streams, by exploiting the spatio-temporal redundancies through a *learned entropy model*. We aim to maximize the reconstruction quality of these point clouds while reducing their joint bitrate. Every point in a LiDAR point cloud contains both a spatial 3D location $(x, y, z)$, as well as an intensity value $r$, and we jointly compress both.

Our method is shown in Fig. 1. We first quantize and encode all point spatial coordinates in the stream into an octree representation, where leaves represent the quantized points and intermediate nodes contain 8-bit symbols representing child occupancies (Sec. 2.1). We then present a novel deep entropy model (Sec. 2.2): a probability model that utilizes *spatio-temporal context* to predict occupancy symbols for each node (Sec. 2.3), as well as intensity values for each point for intensity compression (Sec. 2.4). The outputs of these entropy models are finally fed into a lossless entropy coding algorithm, such as range coding, to produce the final bitstream (Sec. 2.5).

### 2.1 Octree Representation

**Octree Structure and Bit Representation:** LiDAR point clouds are intrinsically challenging to process due to their sparsity and inherently unstructured nature. A tool to counteract these challenges is to use a tree-based data structure, such as an octree or KD-tree, to efficiently partition the space. Inspired by [1, 6], we quantize and represent every point cloud in our stream as an octree with an associated depth value $D$, corresponding to the quantized precision of the point cloud.

Specifically, an octree can be constructed from a 3D point cloud by first partitioning the spatial region into 8 octants, and recursively partitioning each octant until each node contains at most one point, or until $D$ is reached. The resulting octree contains both intermediate nodes and leaf nodes. Each intermediate node can be represented by an 8-bit occupancy symbol $\mathbf{x}$, representing the occupancies of its children; each node also has an implied spatial position. Each leaf node contains one point of the point cloud, and stores the offset between the point and its corner position, as well as the point intensity. We determine the intensity value of each point in the quantized point cloud by taking that of its nearest neighbor in the original point cloud. The number of bits allocated to each leaf node is level-dependent; an octree with $D = k$ will store $k - i$ bits for a leaf node at level $i, i \le k$. Hence,

the octree is memory-efficient—shared bits are encoded with intermediate nodes and residual bits with leaves.

**Serialization:** We serialize the octree into two (uncompressed) bytestreams by traversing the octree in breadth-first order. The first bytestream contains the intermediate node occupancy symbols in breadth-first order, and the second bytestream contains the leaf node offsets/intensities encountered during traversal. Our entropy model focuses primarily on the node occupancies/intensities—we demonstrate in our supplementary materials that leaf offsets do not contain meaningful patterns we can exploit. Hence for subsequent sections we denote $\mathcal{P}^{(t)} = (\mathcal{X}^{(t)}, \mathcal{R}^{(t)})$, where $\mathcal{X}^{(t)} = \{\mathbf{x}_1^{(t)}, ..., \mathbf{x}_{m_t}^{(t)}\}$ is the set of occupancy symbols, and $\mathcal{R}^{(t)} = \{\mathbf{r}_1^{(t)}, ..., \mathbf{r}_{n_t}^{(t)}\}$ is the set of intensities. The serialization is lossless; the only loss comes from $D$-dependent octree quantization. This gives a guarantee on reconstruction quality and allows compression efforts to solely focus on bitrate reduction.

## 2.2 Octree-Based Conditional Entropy Module

The octree sequence is now fed into our *entropy model*. Our entropy model is a probability model that approximates the unknown joint distribution of point clouds $p_{\text{data}}$ with our own distribution $p(\cdot; \mathbf{w})$. Since we convert our point clouds to octree representations, the probability model is equivalent to modeling $p(\mathcal{P}^{(1)}, ..., \mathcal{P}^{(n)}; \mathbf{w})$. According to the classic Shannon's source coding theorem [9], the expected bitrate for the point cloud stream is tightly approximated by the cross-entropy between the real point cloud stream distribution and our parametrized model: $\mathbb{E}_{p_{\text{data}}}[-\log p(\mathcal{P}^{(1)}, ..., \mathcal{P}^{(n)}; \mathbf{w})]$.

We then assume that the joint probability factorizes as follows:

$$\log p(\mathcal{P}^{(1)}, ..., \mathcal{P}^{(n)}; \mathbf{w}) = \sum_t \log p(\mathcal{P}^{(t)}|\mathcal{P}^{(t-1)}; \mathbf{w}) \tag{1}$$

$$= \sum_t \{\log p(\mathcal{X}^{(t)}|\mathcal{P}^{(t-1)}; \mathbf{w}) + \log p(\mathcal{R}^{(t)}|\mathcal{X}^{(t)}, \mathcal{P}^{(t-1)}; \mathbf{w})\} \tag{2}$$

We make a 1st-order Markov assumption: a given octree $\mathcal{P}^{(t)}$ only depends on the sweep preceding it, $\mathcal{P}^{(t-1)}$. We then factor the octree into two entropy models: the node occupancy model $p(\mathcal{X}^{(t)}|\mathcal{P}^{(t-1)}; \mathbf{w})$, and the intensity model $p(\mathcal{R}^{(t)}|\mathcal{X}^{(t)}, \mathcal{P}^{(t-1)}; \mathbf{w})$ conditioned on occupancies. The dependence only on past sweeps makes the model applicable to an online LiDAR stream setting.

## 2.3 Occupancy Entropy Model

We obtain our node occupancy model by continuing to factorize the occupancy probabilities:

$$p(\mathcal{X}^{(t)}|\mathcal{P}^{(t-1)}; \mathbf{w}) = \prod_i p(\mathbf{x}_i^{(t)}|\mathcal{X}_{\text{ans}(i)}^{(t)}, \mathcal{P}^{(t-1)}; \mathbf{w}) \tag{3}$$

Here, $\mathcal{X}_{\text{ans}(i)}^{(t)} = \{\mathbf{x}_{\text{pa}(i)}^{(t)}, \mathbf{x}_{\text{pa}(\text{pa}(i))}^{(t)}, ..., \mathbf{x}_{\text{pa}(...(\text{pa}(i)))}^{(t)}\}$ represents the set of ancestor nodes of $\mathbf{x}_i^{(t)}$ and $\mathcal{P}^{(t-1)}$ represents the point cloud from previous sweep. As seen above, we simplify the autoregressive dependency on ancestors nodes on the octree for the given timestamp, as well as all the nodes at the previous timestamp. We model $p(\cdot|\mathcal{X}_{\text{ans}(i)}^{(t)}, \mathcal{P}^{(t-1)}; \mathbf{w})$ with a deep neural network. The architecture has two backbones, namely the *ancestral node dependence* module which encodes recurrent dependencies on the ancestor nodes $\mathcal{X}_{\text{ans}(i)}^{(t)}$ from the current sweep's octree as well as a *prior octree dependence* module which models information passed from the previous sweep. Fig. 1 depicts the architecture of such network.

**Ancestral Node Dependence:** Our ancestral node dependence module is a recurrent network defined over an ancestral, *top-down* octree path. Inspired by [6], we feed a context feature $\mathbf{c}_i$ for every node $\mathbf{x}_i$ through a multi-layer perceptron (MLP) to extract an initial hidden embedding $\mathbf{h}_{i,0}^{(t)} = \sigma_0(\mathbf{c}_i; \mathbf{w})$, where $\sigma_0(\cdot; \mathbf{w})$ denotes a MLP with learnable parameter $\mathbf{w}$. Context

features include the current octree level, octant spatial location, and parent occupancy; they are known beforehand per node $\mathbf{x}_i$ and computed to facilitate representation learning. We then perform $K_{\text{ans}}$ rounds of aggregation between every node's embedding and its parental embedding: $\mathbf{h}_{i,k}^{(t)} = \sigma_k([\mathbf{h}_{i,k-1}^{(t)}, \mathbf{h}_{\text{pa}(i),k-1}^{(t)}]; \mathbf{w})$. As shorthand, we denote this entire tree-structured recurrent backbone branch as $\mathbf{h}_i^{(t)} = f_{\text{ans}}(\mathbf{x}_i^{(t)}, \mathcal{X}_{\text{ans}(i)}^{(t)})$.

**Temporal Octree Dependence:** We also incorporate the previous octree $\mathcal{P}^{(t-1)}$ into the current entropy model at time $t$ through a *temporal octree dependence* module. We thus first align the previous octree into the sensor coordinate frame of the current octree. Unlike the current octree where we only have access to parental information, we can construct features that make use of *all* information within the previous octree, containing both top-down ancestral information as well as *bottom-up* child information. We exploit this fact by designing a two-stream feature backbone to compute embeddings for every octree node at time $t-1$, inspired by tree-structured message passing algorithms [10, 11]. The forward pass stream is the same as the ancestral dependence module above, generating top-down features from ancestors: $\mathbf{h}_j^{(t-1)} = f_{\text{ans}}(\mathbf{x}_j^{(t-1)}, \mathcal{X}_{\text{ans}(j)}^{(t-1)})$. After the top-down pass, we design a bottom-up aggregation pass, a recurrent network that produces aggregated features from descendants to the current node. Unlike the ancestral module in which each node only has one parent, the number of children per node can vary, and we desire that the output is invariant to the ordering of the children. Hence, we resolve this by designing the following function inspired by deep sets [12]: $\mathbf{g}_j^{(t-1)} = f_{\text{agg},1}(\mathbf{h}_j^{(t-1)} + \sum_{c \in \text{child}(j)} f_{\text{agg},2}(\mathbf{g}_c^{(t-1)}))$, which produces the final aggregated embedding feature containing both top-down and bottom-up context.

**Spatio-Temporal Aggregation:** The final step incorporates the set of aggregated features in the previous octree $\{\mathbf{g}_j^{(t-1)}\}$, with ancestral features in the current octree $\{\mathbf{h}_i^{(t)}\}$ to help with occupancy prediction in the current octree. A key observation is that only a subset of spatially proximal nodes in the previous sweep can contribute to better prediction for a given node at time $t$; moreover, the relative location of each neighbor should define its relative importance. Inspired by this fact, we employ *continuous convolutions* [13] to process previous octree features at the current node. A continuous conv. layer aggregates features from neighboring points to a given node in the following manner: $\mathbf{h}_i = \sum_{j \in \mathcal{N}(i)} \sigma(\mathbf{p}_j - \mathbf{p}_i)\mathbf{h}_j$ where $\mathcal{N}(i)$ is the $i$-th node's $k$-nearest neighboring nodes in 3D space from the $(t-1)$ sweep at the same level as $i$, $\mathbf{p}_i$ is the 3D position of each node, and $\sigma$ denotes a learned MLP. We use a separate MLP with a continuous conv. layer per octree level to process the aggregated features in the previous octree $\{\mathbf{g}_j^{(t-1)}\}_{j \in \mathcal{N}(i)}$ and produce an embedding feature $\mathbf{g}_{i,\text{st}}^{(t)}$.

**Entropy Header:** Finally, the warped feature $\mathbf{g}_{i,\text{st}}^{(t)}$ and ancestral features $\mathbf{h}_i^{(t)}$ are aggregated through a final MLP to output a 256-dimensional softmax of probability values $p(\mathbf{x}_i^{(t)} | \mathcal{X}_{\text{ans}(i)}^{(t)}, \mathcal{P}^{(t-1)}; \mathbf{w})$, corresponding to the predicted 8-bit occupancy for node $i$, time $t$.

## 2.4 Intensity Entropy Model

The goal of the intensity entropy model is to compress extraneous intensities tied to each spatial point coordinate. We assume these intensities are bounded and discrete, so compression is lossless; if they are continuous, there will be a loss incurred through discretization. The model factorizes as follows:

$$p(\mathcal{R}^{(t)} | \mathcal{X}^{(t)}, \mathcal{P}^{(t-1)}; \mathbf{w}) = \prod_i p(\mathbf{r}_i^{(t)} | \mathcal{X}^{(t)}, \mathcal{P}^{(t-1)}; \mathbf{w}) \tag{4}$$

The intent of conditioning on the occupancies $\mathcal{X}^{(t)}$ is not to directly use their values *per se*, but to emphasize that intensity decoding occurs *after* the point spatial coordinates have already been reconstructed in $\mathbb{R}^3$. Therefore, we can directly make use of the spatial position corresponding to each intensity $\mathcal{R}_i^{(t)}$ in compression. We aim to leverage temporal correlations between point intensities across consecutive timestamps to better model the entropy of $\mathbf{r}_i^{(t)}$. Similar to node occupancy prediction above, there is the challenge of how to incorporate previous intensity information when there are no direct correspondences between the two point clouds. We again employ continuous convolutions

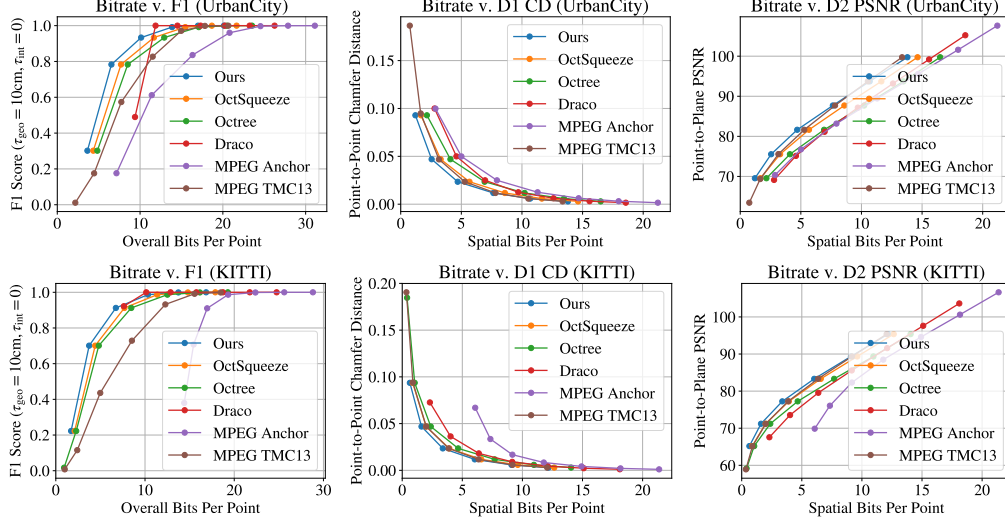

Figure 2: Bitrate *vs.* reconstruction quality curves on UrbanCity (top) and KITTI (bottom). From left-to-right: $F_1$ with $\tau_{\text{geo}} = 10\text{cm}$ and $\tau_{\text{int}} = 0$ ($\uparrow$), point-to-point chamfer distance ($\downarrow$), point-to-plane PSNR ($\uparrow$).

to resolve this challenge. Let $\mathcal{R}_{\mathcal{N}(i)}$ be the set of nearest neighbor intensities $\{\mathbf{r}_j^{(t-1)}\}_{j \in \mathcal{N}(i)}$, where nearest neighbor is defined by spatial proximity of previous point $j$ to the current point $i$. We apply an MLP with a continuous conv. layer that takes the past intensities $\mathbf{r}_j^{(t-1)}$ as input and outputs an embedding feature for each node $i$. This feature is then fed through a linear layer and softmax to output intensity probability values. In our setting we assume our intensity value is an 8-bit integer, so the resulting probability vector is 256-dimensional $p(\mathbf{r}_i^{(t)} | \mathcal{X}^{(t)}, \mathcal{P}^{(t-1)}; \mathbf{w})$.

## 2.5 Entropy Coding Process

**Encoding:** We integrate our entropy model with an entropy coding algorithm (range coding [14]) to produced the final compressed bitstream. During the encoding pass, for every octree in the stream, the entropy model is applied across the octree occupancy bytestream, as well as across the intensities per point, to predict the respective probability distributions. We note that encoding only requires one batch GPU pass per sweep for the occupancy and intensity models. The resulting distributions are then passed to range coding which compresses the occupancies and intensities into two bitstreams.

**Decoding:** The same entropy models are used during decoding. First, the occupancy entropy model is run, given the already decoded past octree, to produce distributions that recover the occupancy serialization and spatial coordinates of the current point cloud. Then, the intensity entropy model is run, given the already decoded intensities in the past point cloud, to produce distributions that recover the current point intensities. Note that our model is well-setup for parallel computation during decoding, for both the occupancies and intensities. As mentioned in Sec. 2.3, the dependence on ancestral nodes instead of all past nodes allows us to only run at most $O(D)$ GPU passes for the occupancy model per sweep. Moreover, the assumed independence between intensities in the current sweep, given the past, allows us to only run 1 GPU pass per sweep for the intensity entropy model.

## 2.6 Learning

Both our occupancy and intensity entropy models are trained end-to-end with cross-entropy loss, for every node $\mathbf{x}_i^{(t)} \in \mathcal{X}_i^{(t)}$ and intensity $\mathbf{r}_i^{(t)} \in \mathcal{R}_i^{(t)}$, for every point cloud in a stream:

$$\ell = \mathbb{E}_{\boldsymbol{P} \sim p_{\text{data}}}\left[-\sum_t \sum_i \log p(\mathbf{x}_{i,gt}^{(t)} | \mathcal{X}_{\text{ans}(i)}^{(t)}, \mathcal{P}^{(t-1)}; \mathbf{w}) - \sum_t \sum_i \log p(\mathbf{r}_{i,gt}^{(t)} | \mathcal{X}^{(t)}, \mathcal{P}^{(t-1)}; \mathbf{w})\right]$$
(5)

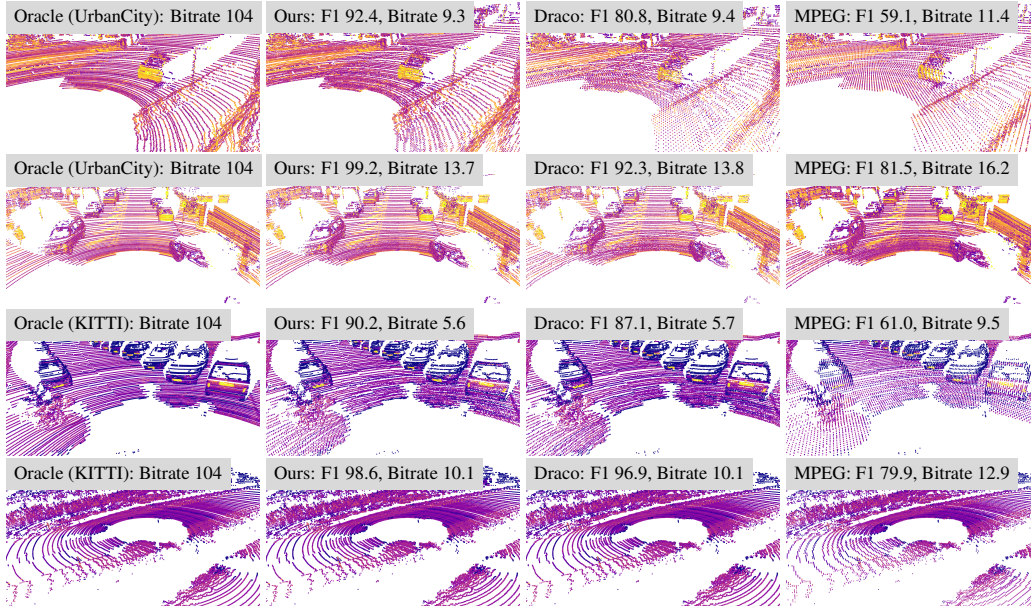

Figure 3: Qualitative results on UrbanCity and KITTI. Points are colored by intensity.

Here, $\mathbf{x}_{i,gt}^{(t)}$ and $\mathbf{r}_{i,gt}^{(t)}$ denote the ground-truth values of the node occupancies/intensities, respectively. As mentioned above, minimizing cross-entropy loss is equivalent to our goal of reducing expected bitrate of the point cloud stream.

## 2.7 Discussion and Related Works

Our approach belongs to a family of point cloud compression algorithms based on tree data structures [15, 2, 16, 17, 18, 19, 20, 1, 20, 21, 22, 23, 24]. Tree-based algorithms are advantageous since they use spatial-partitioning data structures that can efficiently represent sparse and non-uniformly dense 3D point clouds. Two notable examples are Google's Draco [2] and the MPEG anchor [25], which use a KD-tree codec [15] and an octree codec [1] respectively. To exploit temporal redundancies, the MPEG anchor encodes successive point clouds as block-based rigid transformations of previous point clouds; this, however, narrows its usefulness to scenes with limited motion. Moreover, these prior works use simple entropy models that do not fully exploit redundant information hidden in LiDAR point clouds; *e.g.*, repetitive local structures, objects with strong shape priors, *etc*. In contrast, we use a learned entropy model to directly capture these dependencies.

Our approach is also related to work in deep point cloud compression [4, 5, 26, 27, 28, 29, 6]. In particular, both our approach and the prior state-of-the-art [6] use deep entropy models that operate on octree structures directly. However, they do not model temporal redundancies between successive point clouds and compress LiDAR geometry only. In this work, we propose a unified framework that aggregates spatio-temporal context to jointly compress both LiDAR geometry and intensity.

Finally, our work is inspired by recent successes in deep image compression [30, 31, 32, 33, 34, 35, 36] and video compression [37, 38, 39, 40, 41, 42, 43], many of which use deep entropy models.

## 3 Experimental Evaluation

We evaluate our LiDAR compression method on two large-scale datasets. Holding reconstruction quality equal, our framework for joint geometry and intensity compression achieves a 7–17% and 6–19% bitrate reduction over OctSqueeze [6], the prior state-of-the-art in deep point cloud compression, on UrbanCity and SemanticKITTI. Holding bitrate equal, our method's reconstructions also have a smaller realism gap on downstream tasks.

| | | | | Spatial Bitrate | | |
|---|---|---|---|---|---|---|
| O | T | B | CC | D = 12 | D = 14 | D = 16 |
| ✓ | | | | 2.91 | 8.12 | 14.16 |
| ✓ | ✓ | | | 2.87 | 8.04 | 14.08 |
| ✓ | ✓ | ✓ | | 2.72 | 7.90 | 13.95 |
| ✓ | ✓ | ✓ | ✓ | **2.55** | **7.64** | **13.79** |

Table 1: Abalation study of occupancy entropy model on UrbanCity. **O**, **T**, and **B** stand for using past occupancy bytes, top-down aggregated features, and bottom-up aggregated features. **CC** indicates using continuous conv. **D** stands for the octree's max depth.

| | | Intensity Bitrate | | |
|---|---|---|---|---|
| Encoder | P | D = 12 | D = 14 | D = 16 |
| zlib [45] | | 2.42 | 4.79 | 5.23 |
| MLP | ✓ | 2.31 | 4.62 | 5.01 |
| CC | ✓ | **2.13** | **4.30** | **4.68** |

Table 2: Ablation study of intensity entropy model on SemanticKITTI. zlib is an off-the-shelf library [45]; **MLP** is our model without continuous conv.; and **CC** is our final model. **P** stands for using past intensity information. **D** stands for the octree's max depth.

## 3.1 Experimental Details

We validate the performance of our approach on two datasets: UrbanCity [7] and SemanticKITTI [8].

**UrbanCity:** UrbanCity is a large-scale dataset collected by a fleet of self-driving vehicles in several cities across North America [7]. Every sequence consists of 250 Velodyne HDL-64E LiDAR sweeps sampled at 10Hz, each containing a 3D point cloud (as 32-bit floats) and their intensity values (as 8-bit unsigned integers). The average size of each sweep is 80,156 points. We train our entropy models on 5000 sequences and evaluate on a test set of 500. Every sweep in UrbanCity is annotated with per-point semantic labels for the vehicle, pedestrian, motorbike, road, and background classes, as well as bird's eye view bounding boxes for the first three classes. We use these labels to perform downstream perception experiments on the same train/test split.

**SemanticKITTI:** We also conduct compression and downstream perception experiments on SemanticKITTI [8], which enhances the KITTI [44] dataset with dense semantic labels for each LiDAR sweep. It consists of 22 driving sequences containing a total of 43,552 Velodyne HDL-64E LiDAR sweeps sampled at 10Hz. The average size of each sweep is 120,402 points. In our experiments, we use the official train/test splits: sequences 00 to 10 (except for 08) for training and sequences 11 to 21 to evaluate reconstruction quality. Since semantic labels for the test split are unavailable, we evaluate downstream tasks on the validation sequence 08.

**Baselines:** We compare against a number of state-of-the-art LiDAR compression algorithms: Huang *et al.*'s deep octree-based method (**OctSqueeze**) [6], Google's KD-tree based method (**Draco**) [2], Mekuria *et al.*'s octree-based MPEG anchor (**MPEG Anchor**) [1][1], and **MPEG TMC13**[2]. From discussions with the authors, "MPEG Anchor" in [6] is a custom implementation that uses an empirical histogram distribution to compress octree occupancy symbols; we report this baseline as **Octree**. As OctSqueeze and Octree do not compress LiDAR intensities, we augment them with an off-the-shelf lossless compression algorithm [45]. In particular, we first assign an intensity to each encoded point based on the intensity of its nearest neighbour in the original point cloud. Then, we compress the resulting bytestream. For MPEG Anchor, we use the built-in PCL color coder in the authors' implementation, which encodes the average intensity at each leaf node in the octree with range coding. Similarly, for Draco and MPEG TMC13, we use their built-in attributes coders. We also compare against a video compression based algorithm using LiDAR's range image representation (**MPEG Range**). As this baseline was uncompetitive, we report its results in the supplementary.

**Implementation Details:** In our experiments, we construct octrees over a 400m × 400m × 400m region of interest centered on the ego-vehicle. By varying the octree's maximum depth from 11 to 16, we can control our method's bitrate-distortion tradeoff, with spatial quantization errors ranging from 9.75cm (at depth 11) to 0.3cm (at depth 16). We train and evaluate individual entropy models at each depth from 11 to 16, which we found gave the best results. Our models use $K_{ans} = 4$ rounds of aggregation and $k = 5$ nearest neighbors for continuous convolution. Our method is implemented in PyTorch [46] and we use Horovod [47] to distribute training over 16 GPUs. We train our models over 150,000 steps using the Adam optimizer [48] with a learning rate of $1e-4$ and a batch size of 16.

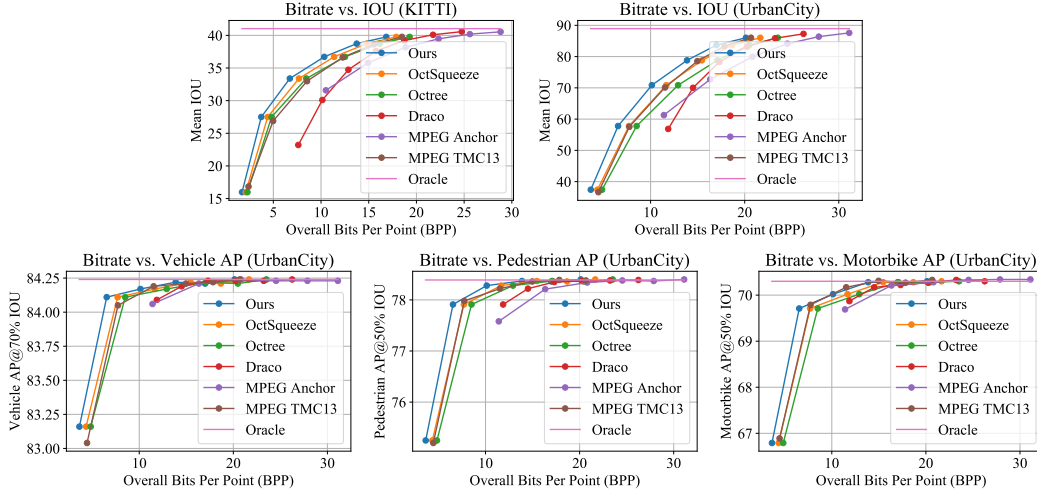

Figure 4: Bitrate *vs*. downstream task performance. Top: mean IoU for semantic segmentation on SemanticKITTI (left) and UrbanCity (right). Bottom: AP for vehicle, pedestrian, and motorbike detection on UrbanCity.

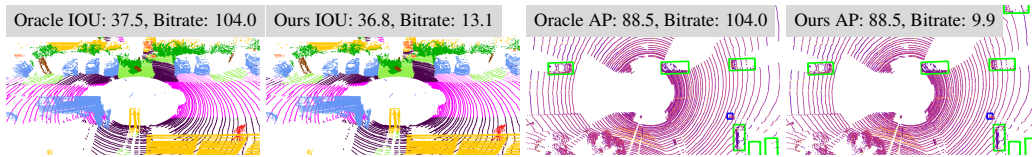

Figure 5: Left: Semantic segmentation on SemanticKITTI. Right: Object detection on UrbanCity. Even at very low bitrates, our reconstructions have a minimal realism gap for downstream tasks.

**Metrics:** We report reconstruction metrics in terms of $F_1$ score, point-to-point (D1) Chamfer distance [6], and point-to-plane (D2) PSNR [49]. Point-to-point and point-to-plane errors are standard MPEG metrics [25]. But whereas they measure reconstruction quality in terms of geometry only, $F_1$ measures this in terms of both geometry and intensity. Reconstruction metrics are averaged across sweeps and bitrate is the average number of bits used to store a LiDAR point. Following standard practice, we do not count the one-time transmission of network weights since it is negligible compared to the size of long LiDAR streams; *e.g*. 1 hour. See our supplementary materials for details.

## 3.2 Results

**Quantitative Results:** In Fig. 2, we report bitrate *vs*. reconstruction quality curves for all competing methods on UrbanCity and SemanticKITTI. The leftmost figures show the trade-off between overall bitrate *vs*. $F_1$. Here, we see that our method outperforms the prior state-of-the-art and, holding reconstruction quality equal, achieves a 7–17% (*resp*., 6–19%) relative reduction in bitrate versus OctSqueeze on UrbanCity (*resp*., SemanticKITTI). Our model also outperforms MPEG TMC13—the MPEG point cloud compression standard—especially at lower bitrates. The right two figures show the trade-off between spatial bitrate *vs*. Chamfer distance and PSNR respectively. Although our method shares a common octree data structure with OctSqueeze (*resp*., Octree), and thus have the same reconstruction quality, we achieve a 5–30% (*resp*., 15–45%) reduction in spatial bitrates on UrbanCity by additionally exploiting temporal information; similar results also hold in SemanticKITTI. These results validate our unified framework for geometry and intensity compression using spatial-temporal information.

**Qualitative Results:** In Fig. 3, we show reconstructions from our method, Draco, and MPEG Anchor on UrbanCity and SemanticKITTI. At similar bitrates, our method yields higher quality reconstructions than the competing methods in terms of both geometry and intensity. For example, from the first and third rows of Fig. 3, we can see that our method produces faithful reconstructions even at high compression rates. In contrast, Draco and MPEG Anchor produce apparent artifacts.

**Ablation Studies:** We perform two ablation studies on our occupancy and intensity entropy models. In Tab. 1, we ablate how to incorporate past information to lower the entropy of our occupancy model. We start with using the past octree's occupancy bytes (**O**) and then progressively add top-down and bottom-up aggregated features (**T** and **B** respectively), and finally continuous convolutions (**CC**). We see that, holding reconstruction quality equal, each aspect of our model consistently reduces bitrates. In Tab. 2, we compare three compression methods for the intensity model: the zlib library, a multi-layer perceptron entropy model (**MLP**), and our final model (**CC**). Note that both MLP and CC conditions on context from neighboring points in the past sweep; zlib does not. However, whereas MLP uses context from one neighbor only, CC aggregates context from multiple neighbors via continuous convolutions. Our results show that learning to incorporate past context reduces intensity bitrates by 4–5%, and that this improvement is strengthened to 11–12% by using continuous convolutions to align information across space and time. Please see our supplementary for details.

**Impact on Downstream Tasks:** To study the impact of compression on downstream perception tasks, we first train segmentation and detection models on uncompressed LiDAR for SemanticKITTI and UrbanCity. Note that these models use both LiDAR geometry and intensity as input (see supplementary for details). Next, we evaluate the models on LiDAR reconstructions obtained from various compression schemes and report their performance as a function of overall bitrate. For segmentation on SemanticKITTI and UrbanCity, we report mean IOU using voxelized ground truth labels at a 10cm resolution. For detection on UrbanCity, we report AP at 50% IOU for pedestrians and motorbikes and 70% IOU for vehicles. In Fig. 4 and 5, we see that our method's reconstructions have the smallest realism gap for downstream tasks across all bitrates. This result is especially pronounced for segmentation models, which are more sensitive to fine-grained geometric and intensity details.

## 4 Conclusion

We have presented a novel LiDAR point cloud compression algorithm using a deep entropy model which exploits spatio-temporal redundancies between successive LiDAR point clouds. We showed that we can compress point clouds at identical reconstruction quality to the state-of-the-art while lowering bitrate significantly, as well as compress LiDAR intensity values effectively which was not as extensively explored by prior works. Furthermore, we showed our compression can be applied to downstream self-driving perception tasks without hindering performance. Looking forward, we plan to extend our method to jointly compress data streams from entire sensor suites.

## Broader Impact

On an immediate level, our contributions are directly applicable as a data compression algorithm in a novel problem setting: the greater we can maximize the performance of such an algorithm, the more we can reduce the storage cost and space required by point clouds. We hope that this in turn unlocks a milestone towards fulfilling our ultimate vision: scaling up the research and deployment of intelligent robots, such as self-driving vehicles, that will revolutionize the safety, efficiency, and convenience of our transportation infrastructure. By capturing the 3D geometry of the scene, LiDAR sensors have proven to be crucial in effective and safe prediction/planning of these robots. Currently, LiDAR sensors are not only expensive due to the upfront cost, but also due to the recurring costs of the massive quantities of data they generate. Good point cloud and LiDAR compression algorithms will thus help to democratize the usage of LiDAR by making it more feasible for people to own and operate. Perhaps just as importantly, our responsibility as researchers in a novel problem area led us to carefully consider the downstream impacts of such a compression algorithm—if the primary usage of LiDAR currently is on perception tasks, such as detection and segmentation, then we need to demonstrate how compression bitrate affects perception performance, helping the community determine the acceptable bitrate at which compression can be used for safe vision and robotics applications. We hope that our work inspires the community to further advance sensor compression in addition to the traditional image and video settings.

## Footnotes

[1]We use the authors' implementation: `https://github.com/cwi-dis/cwi-pcl-codec`.

[2]MPEG TMC13 reference implementation: `https://github.com/MPEGGroup/mpeg-pcc-tmc13`

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
