[Supplementary Material · supp.pdf]

# Supplementary Material — MuSCLE: Multi Sweep Compression of LiDAR using Deep Entropy Models

**Sourav Biswas**[1,2]  **Jerry Liu**[1]  **Kelvin Wong**[1,3]  **Shenlong Wang**[1,3]  **Raquel Urtasun**[1,3]

[1]Uber Advanced Technologies Group  [2]University of Waterloo  [3]University of Toronto

{souravb,jerryl,kelvin.wong,slwang,urtasun}@uber.com

## Abstract

In our supplementary material, we provide additional experimental results that further validate the design and performance of our algorithm (Sec. 1). We first describe additional architecture details for our occupancy and intensity entropy models, as well as their ablation variants (Sec. 2). We also provide additional experimental details with respect to our metrics and downstream perception experiments (Sec. 3). Furthermore, we exhibit an extensive collection of qualitative results in SemanticKITTI and UrbanCity that showcases the performance of our method versus prior state-of-the-art (Sec. 4). Finally, we attach a video (`supp_vid_export.mp4`) that provides an overview of our approach as well as additional qualitative results.

## 1 Additional Experiments

### 1.1 Compression of Leaf Offsets

We mention in Sec. 2.1 of the main paper that we do not attempt to compress the leaf offsets from the octree. The reason is that we experimented with a few compression baselines and were not able to obtain a bitrate improvement over the uncompressed leaf offsets. We experiment with the zlib [1], LZMA [2], and bzip2 [3] compression algorithms on the leaf offset stream from UrbanCity. The results are shown in Tab. 1; we surprisingly found that in all cases the compressed string was longer than the uncompressed one.

|  | Uncompressed | zlib [1] | LZMA [2] | bzip2 [3] |
|---|---|---|---|---|
| Avg. Bytes / Sweep | 102429.31 | 102468.93 | 102493.84 | 103242.28 |

Table 1: Comparison of compression algorithms on leaf offsets from UrbanCity, in terms of average bytes per sweep.

There can be room for future work in entropy modeling the leaf offsets, but our current hypothesis is that since the intermediate octree nodes already encode the shared bits between points, the leaf offsets represent residual bits that can be considered "higher-frequency" artifacts (similar to residual frames in video compression), and are therefore harder to compress.

### 1.2 Using a Range Image Representation

We mention in Sec. 3.1 of the main paper that we designed a range image-based compression baseline. Towards this goal, we first converted point cloud streams in UrbanCity and KITTI into range image representations, which store LiDAR packet data into a 2D matrix. We consider two possible range image representations. The first contains dimensions $H_{\text{lid}} \times W_{\text{azm}}$, where the height dimension represents the separate *laser ID's* of the LiDAR sensor, and the width dimension represents the discretized azimuth bins between -180°and 180°. Each pixel value represents the distance

Figure 1: Bitrate *vs*. reconstruction quality curves on UrbanCity (top) and KITTI (bottom). From left-to-right: $F_1$ with $\tau_{\text{geo}} = 10\text{cm}$ and $\tau_{\text{int}} = 0$ ($\uparrow$), point-to-point chamfer distance ($\downarrow$), point-to-plane PSNR ($\uparrow$).

returned by the laser ID at the specific azimuth angle. Such a representation requires sufficient auxiliary calibration and vehicle information in order to reconstruct the points in Euclidean space—for instance, a separate transform matrix per laser and velocity information to compensate for rolling shutter effects. We use this representation for UrbanCity because we have access to most required information; unfortunately, not every log contains detailed calibration or precise velocity information, requiring us to use approximations.

The second representation simply projects the spatial coordinates of the point cloud sweep into the coordinate frame of the sensor, and does not require a map between laser ID and Euclidean space. Such an image contains dimensions $H_{\text{pitch}} \times W_{\text{azm}}$, where the height dimension now represents discretized pitch angles; each pixel value now represents the distance of a given point from the sensor frame at a given pitch and azimuth bin. We use this representation for our KITTI point clouds, since the dataset does not provide detailed laser calibration information.

We explore both geometry-only and geometry + intensity representations. Spatial positions are encoded in the 8-bit R,G channels of the png image (16 bits total). If intensity is encoded, it is encoded in the B channel. We run H.264 on the png image sequence as our compression algorithm. We evaluate on the same reconstruction metrics: point-to-point Chamfer distance and point-to-plane PSNR (geometry), and $F_1$ score (geometry + intensity).

We show here in Fig. 1, that the results were uncompetitive—the range image representation underperforms other baselines and our approach on every evaluation metric. We observe that even the "lossless" representation (the right-most point on the curves) does not yield perfect reconstruction metrics. This can be surprising for the laser ID representation in UrbanCity. But we hypothesize that the errors come from approximations of the true calibration values (which are not obtainable for every log), as well as the velocity used in rolling shutter compensation—we found that small perturbations in these calibration values yield a large variance in reconstruction quality and metrics.

## 2 Additional Architecture Details

In this section we provide additional architecture details of our octree occupancy and intensity entropy models (Secs. 2.3 and 2.4 in main paper). We also provide architecture details of the models used in the ablation studies of the occupancy and intensity model (Tab. 1, Tab. 2 in main paper).

### 2.1 Occupancy Entropy Model

**Ancestral Node Dependence:** The context feature $\mathbf{c}_i$ consists of the octree level of the current node (1–16), spatial location of the node's octant $(x, y, z)$, octant index of the node relative to its

parent (0–8), and parent occupancy byte (0–255), as well as occupancy byte in the corresponding node in the previous octree (0–255 if exists, 0 otherwise). The initial feature extractor is a 4-layer MLP with fully-connected (fc) layers and intermediate ReLU activations. The hidden layer dimension is 128. Then, every aggregation round consists of a 2-layer fc/ReLU MLP with a 256-dimensional input (concatenating with the ancestor feature), and a hidden dimension of 128. We set the number of aggregation rounds, $K_{\mathrm{ans}}$, to 4.

**Temporal Octree Dependence:** The top-down pass to generate $\mathbf{h}_j^{(t-1)}$ has essentially the same architecture as the ancestral node dependence module above. The one difference is that each context feature additionally includes the "ground-truth" occupancy byte of each node, since each node in sweep $t-1$ has already been decoded. Moreover, each hidden dimension is 64 instead of 128.

Next, recall that the bottom-up aggregation pass has the following formulation:

$$\mathbf{g}_j^{(t-1)} = f_{\mathrm{agg},1}(\mathbf{h}_j^{(t-1)} + \sum_{c \in \mathrm{child}(j)} f_{\mathrm{agg},2}(\mathbf{g}_c^{(t-1)}))$$

Here, $f_{\mathrm{agg},2}$ is a 2-layer fc/ReLU MLP taking a 64-dim input and outputting a 32-dim intermediate embedding. $f_{\mathrm{agg},1}$ is a 2-layer fc/ReLU MLP taking a $(32 + 64)$-dim embedding (child embedding + top-down embedding), and outputting a 64-dim embedding for the current node $j$. The bottom-up pass is run starting from the lowest level $D$ (where there are no children) back up to level 0.

**Spatio-Temporal Aggregation and Entropy Header:** Recall that the continuous convolution layer has the formulation

$$\mathbf{h}_i = \sum_{j \in \mathcal{N}(i)} \sigma(\mathbf{p}_j - \mathbf{p}_i)\mathbf{h}_j$$

where $\mathcal{N}(i)$ is the $i$-th node's $k$-nearest neighbors in sweep $t-1$, at the same octree level as node $i$, and $p_i$ is the 3D position of each node. Here, $\sigma$ is a learned *kernel function*, and it is parameterized by an MLP, inspired by [4]. The MLP contains 3 fc/ReLU layers (no ReLU in last layer), with output dimensions 16, 32, and 64 respectively. The continuous conv layer produces the warped feature $\mathbf{g}_{i,\mathrm{st}}^{(t)}$.

The warped feature $\mathbf{g}_{i,\mathrm{st}}^{(t)}$ and ancestral feature $\mathbf{h}_i^{(t)}$ are aggregated through a final, 4-layer fc/ReLU MLP with hidden dim 128. The prediction header outputs a softmaxed, 256-dim vector of occupancy predictions.

## 2.2 Intensity Entropy Model

The input to the intensity entropy MLP consists of the $k$-nearest neighbor intensities in sweep $t-1$: $\{\mathbf{r}_j^{(t-1)}\}_{j \in \mathcal{N}(i)}$. We set $k = 5$. In addition to the raw intensity value, we include the following features per $\mathbf{r}_j^{(t-1)}$: spatial $(x, y, z)$ position $\in \mathbb{R}^3$, delta vector to current point $\in \mathbb{R}^3$, and 1-D distance value. Hence each point contains an 8-dimensional feature.

Each feature per $\mathbf{r}_j^{(t-1)}$ is then independently given to a 4-layer MLP, consisting of fc layers and ReLU activations. The dimension of each hidden layer is 128. Then, the $k$ output features are input to a continuous convolution layer to produce a single 128-dimensional embedding. The kernel function $\sigma$ of the continuous conv. is parameterized with the same MLP as the one used in spatio-temporal aggregation in the occupancy model. The final predictor is a fc layer and softmax with a 256-dim. output.

## 2.3 Ablation Study Architectures

We first describe the architectures of the occupancy ablation in Tab. 1 of the main paper.

- **O** uses the past occupancy byte to model temporal dependence. The byte is taken from the corresponding node in the previous octree if it exists; if it does not, the feature is zeroed out. This past occupancy byte is then appended to the context feature $\mathbf{c}_i$ (along with parent occupancy byte, octree level, etc.) and fed to the ancestral dependence module. There is no temporal octree dependence module or spatio-temporal aggregation; the final prediction header is directly attached to the ancestral feature.

- **O**, **T** includes the temporal octree dependence module, but removes the bottom-up pass. Hence the final feature produced from this module is $\mathbf{h}_j^{(t-1)}$ (as opposed to $\mathbf{g}_j^{(t-1)}$). There does not exist a spatio-temporal aggregation module using continuous convolutions to produce an embedding for every node $i$. Instead, we use a simpler "exact matching" heuristic similar to including the occupancy bytes—$\mathbf{h}_j^{(t-1)}$ will only be included as a feature for node $i$ in sweep $t$, if node $j$ corresponds to the same octant in sweep $(t-1)$ as node $i$ in sweep $t$. If there is no exact correspondence, the feature is zeroed out.

- **O**, **T**, **B** includes the full temporal octree dependence module, including the bottom-up pass to produce $\mathbf{g}_j^{(t-1)}$. As with the above, we do not include our spatio-temporal aggregation module but rather use the exact matching heuristic to include $\mathbf{g}_j^{(t-1)}$ in the corresponding nodes $i$ in sweep $t$ only if the correspondence exists.

- **O**, **T**, **B**, **CC** includes our full model, including using spatio-temporal aggregation with continuous convolutions to produce an embedding feature for every node $i$.

We now describe the architectures of the intensity ablation in Tab. 2 of the main paper.

- **MLP** only utilizes context from one neighbor in sweep $t-1$. First, the nearest neighbor to node $i$ is obtained in sweep $t-1$. We take the neighbor's corresponding intensity, the delta vector to the current position $\in \mathbb{R}^3$, and 1-D distance value as inputs, and feed it through a 4-layer fc/ReLU MLP and a final softmax predictor head to output 256-dim probabilities.

- **CC** contains the full intensity model with continuous convolutions. For architecture details see 2.2.

## 3 Additional Experiment Details

### 3.1 Reconstruction Metrics

In Sec. 3.3 of the main text, we report reconstruction quality in terms of three metrics: $\mathrm{F}_1$ score, point-to-point Chamfer Distance [5], and point-to-plane PSNR [6]. In the following, we explain each metric in detail. Let $\mathcal{P} = \{(\mathbf{p}_i, r_i)\}_{i=1}^N$ be an input LiDAR point cloud, where each $\mathbf{p}_i \in \mathbb{R}^3$ denotes a point's spatial coordinates and $r_i \in \{0, \ldots, 255\}$ its intensity. Furthermore, let $\hat{\mathcal{P}} = \{(\hat{\mathbf{p}}_j, \hat{\mathbf{r}}_j)\}_{j=1}^M$ be its reconstruction, where $\hat{\mathbf{p}}_j$ and $\hat{\mathbf{r}}_j$ are similarly defined.

Our first metric is an $\mathrm{F}_1$ score that measures reconstruction quality in terms of both geometry and intensity:

$$\mathrm{F}_1(\mathcal{P}, \hat{\mathcal{P}}) = \frac{2 \times \text{\# true positives}}{2 \times \text{\# true positives} + \text{\# false positives} + \text{\# false negatives}} \tag{1}$$

where a reconstructed point $(\hat{\mathbf{p}}_j, \hat{\mathbf{r}}_j) \in \hat{\mathcal{P}}$ is a *true positive* if and only if there exists a point $(\mathbf{p}_i, \mathbf{r}_i) \in \mathcal{P}$ such that $\|\mathbf{p}_i - \hat{\mathbf{p}}_j\|_2 \leq \tau_{\text{geo}}$ and $|\mathbf{r}_i - \hat{\mathbf{r}}_j| \leq \tau_{\text{int}}$. *False positives* are the reconstructed points in $\hat{\mathcal{P}}$ that are not true positives, and *false negatives* are the original points in $\mathcal{P}$ for which no reconstructed point is a true positive. In our experiments, we use $\tau_{\text{geo}} = 10\text{cm}$ and $\tau_{\text{int}} = 0$, and we report $\mathrm{F}_1$ as a function of overall bitrates; *i.e.*, the number of bits to store $\mathbf{p}$ and $\mathbf{r}$. We further report the $\mathrm{F}_1$ score for $\tau_{\text{geo}} \in \{5\text{cm}, 10\text{cm}, 15\text{cm}\}$ and $\tau_{\text{int}} \in \{0, 5, 10\}$ in Fig. 2.

Following the MPEG standards, we also use two standard metrics that measure reconstruction quality in terms of geometry only [7]. We report these metrics as a function of spatial bitrates; *i.e.*, the number of bits to store $\mathbf{p}$. The first such metric measures the point-to-point error between the original point cloud $\mathcal{P}$ and the reconstructed point cloud $\hat{\mathcal{P}}$; this metric is often called the D1 error in the MPEG standards. In our paper, we report this metric as a symmetric Chamfer distance:

$$\mathrm{CD}_{\text{sym}}(\mathcal{P}, \hat{\mathcal{P}}) = \max \left\{ \mathrm{CD}(\mathcal{P}, \hat{\mathcal{P}}), \mathrm{CD}(\hat{\mathcal{P}}, \mathcal{P}) \right\} \tag{2}$$

$$\text{where } \mathrm{CD}(\mathcal{P}, \hat{\mathcal{P}}) = \frac{1}{|\mathcal{P}|} \sum_{\mathbf{p}_i \in \mathcal{P}} \min_{\hat{\mathbf{p}}_j \in \hat{\mathcal{P}}} \|\mathbf{p}_i - \hat{\mathbf{p}}_j\|_2 \tag{3}$$

Figure 2: Bitrate *vs*. $F_1$ curves on UrbanCity (top three rows) and KITTI (bottom three rows). We report $F_1$ across various spatial and intensity thresholds: $\tau_{\text{geo}} \in \{5\text{cm}, 10\text{cm}, 15\text{cm}\}$ and $\tau_{\text{int}} \in \{0, 5, 10\}$.

The second metric measures the point-to-place error between the original point cloud $\mathcal{P}$ and the reconstructed point cloud $\hat{\mathcal{P}}$; this metric is often called the D2 error in the MPEG standards. In our paper, we report this metric in terms of its peak signal-to-noise ratio (PSNR):

$$\mathrm{PSNR}(\mathcal{P}, \hat{\mathcal{P}}) = 10 \log_{10} \frac{3r^2}{\max\{\mathrm{MSE}(\mathcal{P}, \hat{\mathcal{P}}), \mathrm{MSE}(\mathcal{P}, \hat{\mathcal{P}})\}} \tag{4}$$

where $\mathrm{MSE}(\mathcal{P}, \hat{\mathcal{P}}) = \frac{1}{|\mathcal{P}|} \sum_i ((\mathbf{p}_i - \hat{\mathbf{p}}_i) \cdot \hat{\mathbf{n}}_i)^2$ is the mean squared point-to-plane distance, $\hat{\mathbf{n}}_i$ is the normal vector on $\hat{\mathbf{p}}_i$, $\hat{\mathbf{p}}_i = \mathrm{argmin}_{\hat{\mathbf{p}} \in \hat{\mathcal{P}}} \|\mathbf{p}_i - \hat{\mathbf{p}}\|_2^2$ is $\mathbf{p}_i$'s nearest neighbor point in $\hat{\mathcal{P}}$, and $r$ is the peak constant value. We estimate the normal $\boldsymbol{n}_i$ at each point $\mathbf{p}_i \in \mathcal{P}$ using the Open3D function `estimate_normals` with $k = 12$ nearest neighbors [8], and we compute the normal $\hat{\boldsymbol{n}}_i$ corresponding to each point $\hat{\mathbf{p}}_i \in \hat{\mathcal{P}}$ by taking the normal of its nearest neighbor in the original point cloud $\mathcal{P}$. Following the MPEG standard, for each dataset, we compute $r$ as the maximum nearest neighbor distance among all point clouds in the dataset:

$$r = \max_{\mathcal{P}} \max_{\mathbf{p}_i \in \mathcal{P}} \min_{j \neq i} \|\mathbf{p}_i - \mathbf{p}_j\|_2 \tag{5}$$

For UrbanCity, we use $r = 98.69$ and for SemanticKITTI, we use $r = 59.70$.

For completeness, we also report the point-to-point error in terms of its peak signal-to-noise ratio and the point-to-plane error as a symmetric Chamfer distance in Fig. 3.

## 3.2 Downstream Experiment Details

In this section, we provide additional details for our downstream perception experiments.

### 3.2.1 Semantic Segmentation

We use a modified version of the LiDAR semantic segmentation model described in [5].

**Input Representation:** Our model takes as input $T$ bird's eye view ("BEV") occupancy grids of the past $T$ input LiDAR point clouds $\{\mathcal{P}^{(t-T+1)}, \ldots, \mathcal{P}^{(t)}\}$, stacked along the height dimension (*i.e.*, the $z$-axis). By treating the height dimension as multi-dimensional input features, we have a compact input representation on which we can use 2D convolutions [9]. Each voxel in the occupancy grids store the average intensity value of the points occupying its volume, or 0 if it contains no points. We use a region of interest of 160m $\times$ 160m $\times$ 5m centered on the ego-vehicle, $T = 5$ past LiDAR point clouds, and a voxel resolution of $0.15625$cm, yielding an input volume $\boldsymbol{x}$ of size $(T \times Z) \times W \times H = 160 \times 1024 \times 1024$.

**Architecture Details:** Our model architecture consists of two components: (1) a backbone feature extractor; and (2) a semantic segmentation head. The backbone feature extractor $\mathrm{CNN}_{\mathrm{BEV}}$ is a feature pyramid network based on the backbone architecture of [10]:

$$\boldsymbol{f}_{\mathrm{BEV}} = \mathrm{CNN}_{\mathrm{BEV}}(\boldsymbol{x}) \tag{6}$$

where $\boldsymbol{f}_{\mathrm{BEV}} \in \mathbb{R}^{C_{\mathrm{BEV}} \times W/4 \times H/4}$ and $C_{\mathrm{BEV}} = 256$.

The semantic segmentation head $\mathrm{CNN}_{\mathrm{sem}}$ consists of four 2D convolution blocks with 128 hidden channels [1], followed by a $1 \times 1$ convolution layer:

$$\boldsymbol{f}_{\mathrm{sem}} = \mathrm{CNN}_{\mathrm{sem}}(\boldsymbol{f}_{\mathrm{BEV}}) \tag{7}$$

where $\boldsymbol{f}_{\mathrm{sem}} \in \mathbb{R}^{(K \times Z) \times W/4 \times H/4}$ and $K$ is the number of classes plus an additional `ignore` class. To extract per-point predictions, we first reshape $\boldsymbol{f}_{\mathrm{sem}}$ into a $K \times Z \times W/4 \times H/4$ logits tensor, then use trilinear interpolation to extract per-point $K$-dimensional logits, and finally apply softmax.

**Training Details:** We use the cross-entropy loss to train our semantic segmentation model. For SemanticKITTI, we follow [12] and reweight the loss at each point by the inverse of the frequency of its ground truth class; this helps to counteract the effects of severe class imbalance. Moreover, we use data augmentation by randomly scaling the point cloud by $s \sim \mathrm{Uniform}(0.95, 1.05)$, rotating it by $\theta \sim \mathrm{Uniform}(-\pi/4, \pi/4)$, and reflecting it along the $x$ and $y$-axes. We use the Adam optimizer [13] with a learning rate of $4\mathrm{e}{-4}$ and a batch size of 12, and we train until convergence.

Figure 3: Bitrate *vs.* reconstruction curves on UrbanCity (top two rows) and KITTI (bottom two rows). We report point-to-point (D1) and point-to-plane (D2) errors in terms of Chamfer distances (left) and PSNR (right).

### 3.2.2   Object Detection

We use a modified version of the LiDAR object detection model described in [5]. It largely follows the same architecture as our semantic segmentation model, with a few modifications to adapt it for object detection. We describe these modifications below.

**Architecture Details:**   Our object detection model consists of two components: (1) a backbone feature extractor; and (2) an object detection head. The backbone feature extractor here shares an identical architecture to that of the semantic segmentation model. The object detection head consists of four 2D convolution blocks with 128 hidden channels followed by a $1 \times 1$ convolution layer to predict a bounding box $\boldsymbol{b}_{i,k}$ and detection score $\alpha_{i,k}$ for every BEV pixel $i$ and class $k$. Each bounding box $\boldsymbol{b}_{i,k}$ is parameterized by $(\Delta x, \Delta y, \log w, \log h, \sin \theta, \cos \theta)$, where $(\Delta x, \Delta y)$ are the position offsets to the object's center, $(w, h)$ are the width and height of its bounding box, and $\theta$ is its heading angle. To remove duplicate bounding boxes predictions, we use non-maximum suppression.

**Training Details:**   We use a combination of classification and regression losses to train our detection model. In particular, for object classification, we use a binary cross-entropy loss with online hard negative mining, where positive and negative BEV pixels are determined based on their distance to an object center [14]. For bounding box regression, we use a smooth $\ell_1$ loss on

$\Delta x, \Delta y, \log w, \log h, \sin \theta, \cos \theta$. We use the Adam optimizer [13] with a learning rate of $4\mathrm{e}{-}4$ and a batch size of 12, and we train until convergence.

## 4 Additional Qualitative Results

In Fig. 4 and 5, we compare the reconstruction quality of our method versus Draco [15] and MPEG anchor [16]. Then, in Figs. 6, 7, and 8, we visualize results from semantic segmentation and object detection on SemanticKITTI and UrbanCity. As shown in these figures, our compression algorithm yields the best reconstruction quality at comparable or lower bitrates than the competing methods.

## Footnotes

[1] Each 2D convolution block consists of a $3 \times 3$ convolution, GroupNorm [11], and ReLU.

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

Oracle (UrbanCity): Bitrate 104.0

Ours: F1 100.0 Bitrate 16.5

Draco: F1 100.0 Bitrate 20.0

MPEG: F1 94.6 Bitrate 20.6

Oracle (UrbanCity): Bitrate 104.0

Ours: F1 93.7 Bitrate 10.8

Draco: F1 93.3 Bitrate 13.9

MPEG: F1 76.6 Bitrate 14.8

Oracle (UrbanCity): Bitrate 104.0

Ours: F1 92.9 Bitrate 10.0

Draco: F1 85.1 Bitrate 10.9

MPEG: F1 53.4 Bitrate 10.4

Figure 4: Qualitative results on UrbanCity. Points are colored by intensity.

Oracle (KITTI): Bitrate 104.0

Ours: F1 100.0 Bitrate 16.4

Draco: F1 100.0 Bitrate 17.0

MPEG: F1 96.1 Bitrate 16.8

Oracle (KITTI): Bitrate 104.0

Ours: F1 91.4 Bitrate 6.6

Draco: F1 90.6 Bitrate 6.8

MPEG: F1 70.2 Bitrate 11.6

Oracle (KITTI): Bitrate 104.0

Ours: F1 90.8 Bitrate 5.6

Draco: F1 89.2 Bitrate 5.8

MPEG: F1 69.2 Bitrate 11.0

Figure 5: Qualitative results on SemanticKITTI. Points are colored by intensity.

Oracle (KITTI): IOU 36.1 Bitrate 104.0

Ours: IOU 34.8 Bitrate 13.8

Draco: IOU 34.7 Bitrate 14.4

MPEG: IOU 31.4 Bitrate 14.5

Oracle (KITTI): IOU 31.3 Bitrate 104.0

Ours: IOU 29.5 Bitrate 6.7

Draco: IOU 29.0 Bitrate 8.4

MPEG: IOU 26.3 Bitrate 13.0

Oracle (KITTI): IOU 37.3 Bitrate 104.0

Ours: IOU 30.6 Bitrate 5.9

Draco: IOU 30.2 Bitrate 6.0

MPEG: IOU 27.6 Bitrate 10.3

Figure 6: Semantic segmentation results on SemanticKITTI. IOU is averaged over all classes.

Figure 7: Semantic segmentation results on UrbanCity. IOU is averaged over all classes.

Figure 8: Object detection results on UrbanCity. AP is averaged over the vehicle, pedestrian, and motorbike classes.