[Reviews · NeurIPS 2020]

Review 1

Summary and Contributions: This paper proposes a new compression method for point cloud data, especially focusing on the point cloud time series obtained by LiDAR. The proposed compression method builds a neural network that models point occupancy on octree. The proposed model explicitly considers dependency among tree nodes and temporal dependency of the two temporally adjacent point clouds. Temporal dependency is modeled as a conditional probabilistic model based on a 1st order Markov assumption. The experimental results show that the proposed techniques realize the better compression-accuracy trade-off, compared with baselines.

Strengths: - Modeling temporal dependency for LiDAR stream data is a simple but promising idea for this task. - The experiments show a substantial improvement over baseline methods.

Weaknesses: Although the proposed method is a good application of neural networks, it is a combination of techniques that are well-known in the NeurIPS community. In this sense, the technical novelty and depth are limited. [AFTER REBUTTAL] Based on the other reviewers' comments and discussion on the contribution, I change my score.

Correctness: - The proposed method correctly models what the authors argued in the introduction. - The experiments were mostly conducted correctly except for some unclear points (see comments below).

Clarity: The paper is basically well-written and easy to follow, although there are some unclear points. Please see my comments below.

Relation to Prior Work: Differences from prior work is clearly mentioned: (1) consideration of temporal dependency, and (2) intensity compression.

Reproducibility: Yes

Additional Feedback: - Is \sigma at Line 138 an MLP? Or does it involve window functions? - Please clarify the definition of bitrate. I think that the authors does not count the storage for the parameters of trained neural network. Is this right? - Also, if I understand correctly, all the evaluations of compression performance include intensities. How about the compression performance if intensities are excluded from evaluation? (because zlib is a general-purpose compressor and it is not very strong baseline here)


Review 2

Summary and Contributions: In this paper, a point cloud compression approach is proposed by using spatial-temporal information. The main contribution comes from the investigation of using temporal information.

Strengths: - The proposed approach is straightforward and effective. - The experimental results are comprehensive. - Demonstrate the effectiveness of compression in the downstream task is very interesting.

Weaknesses: - The paper builds upon previous work OctSqueeze, the novelty may be limited by adding temporal information only. I am not sure whether it is sufficient enough for NeurIPS. - Considering this paper focus on the compression of multiple sweeps of the point cloud, it is recommended to compare with the V-PCC, i.e., TMC2 in the experimental part. - Besides, I am curious why TMC13 is not included in the baseline method. In my opinion, TMC13 is a strong baseline and should be included in the comparison.

Correctness: - The claims are correct and the evaluation results are extensive.

Clarity: This paper is well written and is easy to follow.

Relation to Prior Work: The paper has provide the sufficient discussion with previous work.

Reproducibility: Yes

Additional Feedback: - The computational complexity. Please provide the running time on GPU and CPU platforms, respectively. Besides, it is also required to provide the running time of traditional codecs. - The proposed is trained by using 16GPUs, so please provide the memory consumption of a single batch. Besides, is it possible train the whole model with fewer GPUs? - Please provide the average number of points for testing datasets. - It seems the improvements over previous OctSqueeze baseline will drop at the low-bitrate setting in Fig.2.


Review 3

Summary and Contributions: Edit: I read the rebuttal and continue to support my positive review (7: good submission; accept). This paper presents a method for compressing lidar data including both 3D location and intensity values. The approach builds on existing learning-based lidar compression models that use an octet representation. It expands on those methods by conditioning on the previous lidar sweep, using "continuous convolution" to better model data that doesn't fall on a grid, and by compressing intensity values. These enhancements lead to better compression rates (up to 35%).

Strengths: 1) Lidar compression is an important problem for the ML community that is interesting academically and for real-world applications (self-driving cars being a primary example). 2) The compression gains are impressive. On standard data sets, the proposed method saves between 7% and 35% at equal quality levels. For reference, for video compression, papers are often published that provide a 2% gain, albeit over much more mature baselines, and new standard are typically built when gains reach 25-30%. 3) Analysis and feature aggregation when the data is irregular and doesn't fall on a grid is difficult to do well since the assumptions of standard approaches (conv nets and MLPs) don't fit. Although this paper builds on existing methods ("continuous convolution" and "deep sets"), it adapts them to the 3D octet structure used to represent lidar data. 4) Evaluation is performed both on direct metrics (F1, PSNR, etc.) and on real downstream tasks.

Weaknesses: The comments about the number of GPU passes needed for the occupancy model was not clear to me (Section 2.5). Some extra information would be helpful. Runtime numbers would also strengthen the paper since real-time encoding is needed for live applications.

Correctness: yes

Clarity: yes

Relation to Prior Work: yes

Reproducibility: No

Additional Feedback: Regarding reproducibility, I think additional implementation details would be needed to reproduce the results, but I do *not* fault the authors for not including them in the paper. I think the structure and top priorities (background, related work, the new method, and evaluation) is very appropriate for an 8-page paper. Additional information is also given in the supplementary material.


Review 4

Summary and Contributions: The paper proposes a principled approach to modeling point cloud streams and uses the model for entropy coding of such streams. Resulting compression rates appear to beat the state of the art. Paper is mostly easy to read, well motivated and uses reasonable and elegant formulations.

Strengths: First off, results beat the state of the art. (I have not read other work on Lidar stream compression, so I have to go by comparisons in the paper.) Second, the paper is of the type that inspires further thought, which then means that perhaps the impact will be broader than it might seem. Good probability density models are useful for compression, but are also useful for inference: could these models be used directly for downstream tasks like tracking or segmentation? Also, the way the model is set up, it seems to me that it does not preclude the possibility that the model learns scale-invariant features, in almost a fractal view of the world, where the dependencies at different depths depend not on the depth so much as on occupancies of the nearby nodes, thus allowing the model to generalize what is learned about large objects to smaller objects. (A range of experiments can be done along these lines, from investigating the learned parameters to doing ablation studies, like, can the model's depth be directly increased without retraining the network by duplicating layers from the coarser levels; can the same model be retrofitted to work on higher density clouds, and interestingly, how invariant is it to scaling (shrinking or expanding the point cloud) )

Weaknesses: Couple of small issues: This sentence is unclear: "We obtain intensities in the quantized point cloud by taking the nearest neighbor in the source point cloud." Figure 1 contains a type of an illustration that is typical of modern ML papers (and often required by reviewers). But i find these illustrations to be hand-wavy and I (think I) understood the method from the writing, not from the figure. But that's may be the matter of taste. It seems that 't' variable was never defined (though one could infer its meaning). The context variable 'c' would also be better defined before it is used, rather than after. The range encoding reference format seems to be off. At a higher level, while the paper does compare with the state of the art, the formulation before getting into neural nets suggests possibility of using simpler more direct statistical models. The paper would be stronger if the neural networks could be shown to beat these and how they do it. And, as I mentioned in the strengths as well, discussing the probability model and its strength could lead to a discussion of the model's use beyond compression. Finally, some of the analysis I mentioned above in the strengths would make the paper even more exciting.

Correctness: Seems right.

Clarity: The paper was a joy to read. Harkens back to early NIPS when information theory and ML where more closely related.

Relation to Prior Work: I think the paper is well positioned in the background literature.

Reproducibility: Yes

Additional Feedback:

[Author Response · NeurIPS 2020]

We thank the reviewers for their valuable comments. In the following, we will address each comment in detail.

**R1, R2: Novelty.** To our knowledge, learning-based density models of point cloud sweeps [spatial+intensity] for compression are not widely explored, and modeling temporal relationships between point clouds is non-trivial. Our novel tree-structured density models combine existing techniques (deep sets, cont. convs) to significantly advance beyond the scope of OctSqueeze – fully exploiting temporal redundancies and modeling intensity attributes.

**R2: Additional baselines.** We benchmarked MPEG TMC13 on UrbanCity and SemanticKITTI (see Fig. 1). On average, our approach achieves a better bitrate vs. reconstruction trade-off, especially at lower bitrates. We could not benchmark MPEG TMC2 due to `segmentation fault` errors in their reference implementation [1]. We have emailed the authors and hope to include TMC2 in our final paper. That said, TMC2 is related to our MPEG Range baseline: both compress 2D projections of the point cloud using video codecs. We found that tree-based codecs perform better.

Figure 1: Bitrate vs. reconstruction quality on UrbanCity and SemanticKITTI.

**R2, R3: Runtime/GPU passes.** We benchmarked our approach on a workstation with an Nvidia GTX 1080Ti GPU and Intel Xeon E5 CPU. Our octree data structure and range coder is written in C++ for CPU execution, and our spatial/intensity entropy models run on the GPU. During encoding, we predict distributions over occupancy symbols with one forward pass of the entropy models. During decoding, we require $O(D)$ forward passes ($D$ is the octree's depth), each interleaved with range decoding to reconstruct the octree level by level. We also benchmarked OctSqueeze, Draco, MPEG Anchor, and MPEG TMC13. Our encoding speed is fast, and our total runtime is on par with TMC13. Moreover, there are many opportunities to speed up our research code significantly, especially in CPU/GPU I/O.

| Depth | MuSCLE (Spatial) Encoding (ms) | MuSCLE (Spatial) Decoding (ms) | MuSCLE (Intensity) Encoding (ms) | MuSCLE (Intensity) Decoding (ms) | OctSqueeze (Spatial) Encoding (ms) | OctSqueeze (Spatial) Decoding (ms) | Draco (Spatial) Encoding (ms) | Draco (Spatial) Decoding (ms) | MPEG Achor (Spatial + Intensity) Encoding (ms) | MPEG Achor (Spatial + Intensity) Decoding (ms) | MPEG TMC13 (Spatial + Intensity) Encoding (ms) | MPEG TMC13 (Spatial + Intensity) Decoding (ms) |
|---|---|---|---|---|---|---|---|---|---|---|---|---|
| 11 | 65.6 | 279.5 | 20.8 | 49.0 | 16.5 | 93.0 | 25.8 | 15.3 | 59.6 | 33.6 | 1219.0 | 272.0 |
| 12 | 94.4 | 477.5 | 31.4 | 89.9 | 26.2 | 165.7 | 29.0 | 16.2 | 84.2 | 50.3 | 1906.9 | 461.4 |
| 13 | 135.6 | 764.0 | 42.7 | 137.9 | 40.1 | 299.4 | 30.3 | 17.2 | 106.5 | 67.2 | 2539.0 | 674.4 |
| 14 | 177.1 | 1084.2 | 49.2 | 162.6 | 53.4 | 486.4 | 30.7 | 17.5 | 125.5 | 84.1 | 3038.9 | 839.1 |
| 15 | 202.4 | 1352.1 | 50.7 | 166.3 | 63.6 | 699.0 | 30.9 | 17.6 | 143.0 | 98.9 | 3406.3 | 925.8 |
| 16 | 208.4 | 1569.4 | 55.5 | 165.6 | 65.4 | 902.3 | 31.1 | 17.6 | 160.2 | 112.8 | 3695.1 | 985.0 |

Table 1: Encoding/decoding time (ms) of MuSCLE spatial intensity model, OctSqueeze, Draco, MPEG Anchor, and MPEG TMC13.

**R2, R3: Experiment details.** The average number of points per LiDAR point cloud is 80,156 in UrbanCity and 120,402 in SemanticKITTI. During training, our full model requires 6-10GB of GPU memory. 16 GPUs was our starting point, but we retrained models with 4 GPUs and found roughly equal convergence / test metrics.

**R1: Clarification of line 138 and bitrates.** The $\sigma: \mathbb{R}^3 \to \mathbb{R}^d$ in line 138 is an MLP that predicts $d$-dimensional weights for continuous convolution. Bitrate is the total number of bits used to store the LiDAR dataset divided by the total number of points it contains. We do not count the one-time transmission of network weights (standard practice in learned compression) since this is negligible compared to the size of long LiDAR streams, e.g., 1 hour. We emphasize that the middle and right columns of Fig. 2 depict spatial-only bitrates since they compare spatial reconstruction quality.

**R4: Typos and clarification of line 71.** Thanks, we will fix the typos. Regarding line 71, our quantization scheme may produce a point cloud with fewer points than the original point cloud. Thus, we determine the intensity value of each point in the quantized point cloud by taking that of its nearest neighbour in the original point cloud.

**R4: Simpler entropy models.** Our tree-based baselines use a spectrum of statistical models with varying complexity. For example, Octree and MPEG Anchor use an empirical histogram-based entropy model, MPEG TMC13 uses a cascade of adaptive frequency tables, and OctSqueeze uses a deep learning model.

**R4: Invariance to scale, density, and tree depth.** Thanks for the exciting insight! We agree that ideally our density estimation model based on tree-structured message passing and continuous convolutions should be invariant to scale, density, and tree depth, echoing R4's thoughts on potential applications to other perception tasks. However, in our current implementation, several mechanisms make this not the case: 1) our input feature include information on the depth; 2) we use separate continuous convolution modules per depth; and 3) we do not use data augmentation (e.g., random scaling) and so our network learns the inductive bias of the dataset (e.g., the real-world scale). That said, by modifying these mechanisms, we should get a density model that is invariant to scale, density, and tree depth.

## Footnotes

[1]MPEG TMC13: `github.com/MPEGGroup/mpeg-pcc-tmc13`, MPEG TMC2: `github.com/MPEGGroup/mpeg-pcc-tmc2`.


[Meta-Review · NeurIPS 2020]

Reviewers felt that this is a strong submission with a relatively simple idea that produces a substantial improvement over baseline methods. The paper is fairly well-written. The experimental results are comprehensive, and reviewers especially appreciated the experiments showing the impact of compression on downstream tasks. Reviewers felt that the proposed method is clever and interesting. There was some concern about the technical novelty in relation to prior work (e.g. OctSqueeze), as well as some missing baseline comparisons. There were also questions about the number of forward passes, runtime, GPU memory, etc.